# The immune checkpoint B7x expands tumor-infiltrating Tregs and promotes resistance to anti-CTLA-4 therapy

Peter John[1], Marc C. Pulanco[1], Phillip M. Galbo Jr.[2], Yao Wei[1], Kim C. Ohaegbulam[1], Deyou Zheng [2,3] & Xingxing Zang [1,4,5✉]

Immune checkpoint molecules play critical roles in regulating the anti-tumor immune response, and tumor cells often exploit these pathways to inhibit and evade the immune system. The B7-family immune checkpoint B7x is widely expressed in a broad variety of cancer types, and is generally associated with advanced disease progression and poorer clinical outcomes, but the underlying mechanisms are unclear. Here, we show that trans-duction and stable expression of B7x in multiple syngeneic tumor models leads to the expansion of immunosuppressive regulatory T cells (Tregs). Mechanistically, B7x does not cause increased proliferation of Tregs in tumors, but instead promotes the conversion of conventional CD4[+] T cells into Tregs. Further, we find that B7x induces global transcriptomic changes in Tregs, driving these cells to adopt an activated and suppressive phenotype. B7x increases the expression of the Treg-specific transcription factor Foxp3 in CD4[+] T cells by modulating the Akt/Foxo pathway. B7x-mediated regulation of Tregs reduces the efficacy of anti-CTLA-4 treatment, a therapeutic that partially relies on Treg-depletion. However, combination treatment of anti-B7x and anti-CTLA-4 leads to synergistic therapeutic efficacy and overcomes the B7x-mediated resistance to anti-CTLA-4. Altogether, B7x mediates an immunosuppressive Treg-promoting pathway within tumors and is a promising candidate for combination immunotherapy.

[1] Department of Microbiology & Immunology, Albert Einstein College of Medicine, Bronx, NY, United States. [2] Department of Genetics, Albert Einstein College of Medicine, Bronx, NY, USA. [3] Departments of Neurology and Neuroscience, Albert Einstein College of Medicine, Bronx, NY, United States. [4] Department of Medicine, Albert Einstein College of Medicine, Bronx, NY, United States. [5] Department of Urology, Albert Einstein College of Medicine, Bronx, NY, United States. ✉email: xingxing.zang@einsteinmed.edu

The B7 and CD28 protein families of ligands and receptors mediate critical pathways called immune checkpoints that regulate the activation and function of immune cells, particularly T cells[1]. Immune checkpoints have great relevance to the tumor microenvironment since tumor cells use inhibitory pathways to suppress anti-tumor T cells and thereby evade immune-mediated killing. Tumor cells can express checkpoint ligands themselves, thereby inhibiting nearby effector immune cells and conferring an immunosuppressive tumor microenvironment[2]. Immune checkpoint blockade (ICB) with monoclonal antibodies has proven to be an effective therapeutic strategy for cancer treatment. In particular, antibodies targeting the checkpoints CTLA-4, PD-1, and PD-L1 are effective against a variety of cancer types, establishing the principle that blocking inhibitory immune pathways can promote an anti-tumor immune response and lead to clinical benefit[3].

Despite the success of ICB, resistance to ICB is very common[4,5], suggesting that other pathways beyond the ones currently targeted must significantly contribute to immune evasion in tumors. For example, anti-PD-1/anti-PD-L1 agents have response rates lower than 20% in non-small cell lung cancer (NSCLC)[6]. This suggests alternate immune evasion mechanisms must be active in non-responding tumors. Indeed, PD-L1$^+$ tumors constitute only 25% to 31% of tumors in NSCLC, whereas the majority of tumors are positive for other B7-family checkpoint ligands such as B7x or HHLA2[7]. Similarly, expression of alternate immune checkpoint proteins is associated with resistance to anti-CTLA-4 treatment in prostate cancers[8], which also commonly express checkpoint ligands such as B7x or B7-H3[9]. Therefore, identification and characterization of the other immune checkpoints expressed in tumors is of paramount importance if we are to improve our treatment strategies.

B7x (also called B7-H4, B7S1, or Vtcn1) is a B7-family ligand that has suppressive effects on T cells through a yet-unidentified receptor[10]. In vitro, it directly inhibits the proliferation, cytokine production, and cytotoxic functions of effector T cells[11–13]. B7x has limited expression in normal human tissues[11,14] but is frequently overexpressed in a broad variety of human cancers[9,15–21]. In general, the expression of B7x in tumors is associated with greater disease progression and poorer prognosis[10]. Based on our recent work in murine models, tumor-expressed B7x inhibits the inflammatory functions and promotes exhaustion of effector CD8$^+$ T cells[6,22]. Interestingly, B7x-mediated inhibition of effector T cells also correlates with increased presence of the immunosuppressive population of Foxp3$^+$ CD4$^+$ T cells called regulatory T cells (Tregs). Tregs are a potently suppressive population of T cells that are commonly found in solid tumors and are implicated in maintaining a tolerogenic environment that promotes tumor growth[23]. However, how B7x regulates Treg populations in tumors is not well understood.

Here, we investigate the mechanisms by which B7x regulates the infiltration of Tregs in the tumor microenvironment. With a combination of in vivo and in vitro approaches, we show that B7x expands tumor-infiltrating Treg populations by inducing the expression of the Treg-specific transcription factor Foxp3 in conventional CD4$^+$ T cells and converting them into Tregs. Further, we demonstrate that B7x-mediated Treg expansion reduces the efficacy of anti-CTLA-4 therapy, a treatment that relies partially on Treg-depletion. To overcome this resistance to therapy, we use anti-B7x antibody treatment in combination with anti-CTLA-4 which demonstrate synergistic efficacy. Altogether, we describe here a suppressive pathway of B7x in the tumor microenvironment and demonstrate a promising combination therapy.

## Results

**B7x promotes Tregs within the tumor microenvironment.** To investigate the role of tumor-expressed B7x on Tregs in vivo, we utilized multiple syngeneic murine tumor models (Fig. 1a, Supplementary Fig. 1a). Whereas human tumor cell lines express B7x at high levels, murine cell lines commonly used in syngeneic tumor models do not natively express B7x (Supplementary Fig. 2a, b), therefore we transduced the well-characterized murine tumor cell lines MC38, CT26, and Hepa1-6 to express mouse B7x (Fig. 1b, Supplementary Fig. 1b, e). These stably transduced cell lines express B7x at levels comparable to human cell lines that natively express human B7x (Supplementary Fig. 2b). Cell lines mock-transduced with the empty vector served as the respective B7x$^-$ Control cell lines.

MC38-B7x and MC38-Control cells were engrafted subcutaneously into mouse flanks and were subsequently dissociated for analysis of the CD4$^+$ Foxp3$^+$ Treg population (Fig. 1a–g). The expression of B7x increased Treg populations within these tumors, both in absolute numbers and in relative proportions of CD45$^+$ immune cells (Fig. 1c, d). Notably, the ratio of Treg populations in B7x$^+$ MC38 tumors was elevated in comparison to the Foxp3$^-$ conventional CD4$^+$ T cell and CD8$^+$ T cell populations, indicating that B7x was shifting the immune landscape of the tumor to a more suppressive milieu (Fig. 1d). Greater Treg populations as percentage of CD45$^+$ cells were also observed in the draining lymph nodes of the tumor-bearing mice, although the effect was weaker as compared to the tumor-infiltrating lymphocytes and the Treg: CD8$^+$ T cell ratio was not affected in the lymph nodes (Fig. 1e). Upon analyzing the functional state of the tumor-infiltrating Tregs, we observed that Tregs in B7x$^+$ tumors had greater expression of TGF-LAP (Fig. 1f), a surface marker of TGFβ1 production, suggesting that these cells had greater suppressive capacity than their counterparts in B7x$^-$ tumors. Interestingly, we observed lower expression of the proliferation marker Ki67 in B7x-associated Tregs (Fig. 1f), a surprising finding given the larger Treg populations in B7x$^+$ tumors. Therefore, we hypothesized that the expanded Treg populations did not originate from increased proliferation of pre-existing Tregs but were instead peripherally converted from conventional CD4$^+$ T cells.

In general, Tregs can develop directly from the thymus alongside other T cells (natural Tregs or nTregs), or they can be peripherally converted from conventional CD4$^+$ T cells (peripheral Tregs or pTregs). We sought to identify the origin of the tumor-infiltrating Tregs using lineage markers to determine which population of Tregs B7x acts upon. Neuropilin-1 (Nrp1) is a membrane-bound protein that has been proposed to be associated with thymic-derived nTregs[24]. We observed that Tregs in lymphoid organs such as the spleen and lymph nodes had heterogeneous expression of Nrp1, suggesting a mixture of nTregs and pTregs (Supplementary Fig. 2c). Within tumors, however, Tregs were almost entirely Nrp1$^-$, suggesting that MC38 tumors were heavily infiltrated with peripherally converted Tregs (Supplementary Fig. 2c). We also examined the expression of Helios, a transcription factor initially proposed to be a marker of nTregs, but is now better understood to be a functional marker of activated Tregs[25,26]. In contrast to Nrp1, tumor-associated Tregs were universally Helios$^+$. Thus, both MC38-B7x and MC38-Control tumors were homogenously populated by Nrp1$^-$ Helios$^+$ cells (Fig. 1g). However, since Nrp1 and Helios are not reliable in vivo markers of nTreg/pTreg origin[27], we further dissected the mechanisms by which B7x regulates Tregs in vitro (Fig. 2).

To determine if the effects of B7x on Treg populations is consistent in multiple tumor models, we repeated these studies in the Hepa1-6 and CT26 tumor models as well (Supplementary Fig. 1a–g), and consistently observed increased Treg infiltration in B7x$^+$ tumors as compared to their B7x$^-$ counterparts, as well as

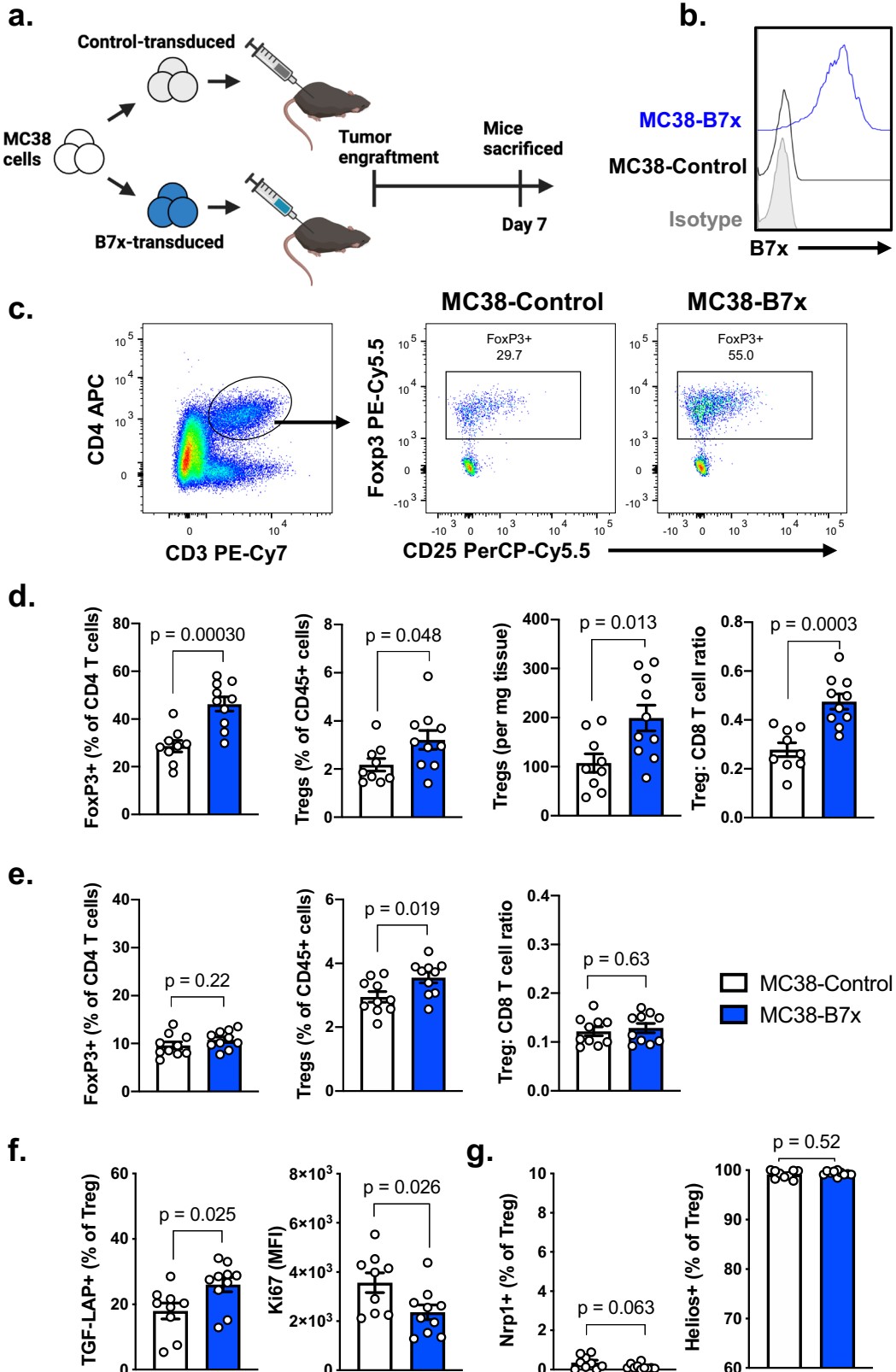

**Fig. 1 Tumor-expressed B7x promotes infiltrating Treg populations. a** Experimental scheme of MC38 tumor engraftment. **b** Representative flow cytometry analysis of B7x expression in MC38-B7x and MC38-Control cells. **c** Representative gating strategy for Tregs in CD45+ cells. **d, e** Populations of CD4+ T cells and CD8+ T cells were analyzed within dissociated tumors (**d**) and draining lymph nodes (**e**). **f, g** Gated on Tregs, surface TGF-LAP and intracellular Ki67 (**f**), and surface Nrp1 and intracellular Helios (**g**) was determined. $n = 9$ mice in control group and 10 mice in B7x group, representative of three independent experiments. For (**d–g**), error bars represent SEM and $P$ values were calculated by two-tailed Student's $T$-test, unadjusted for multiple comparisons.

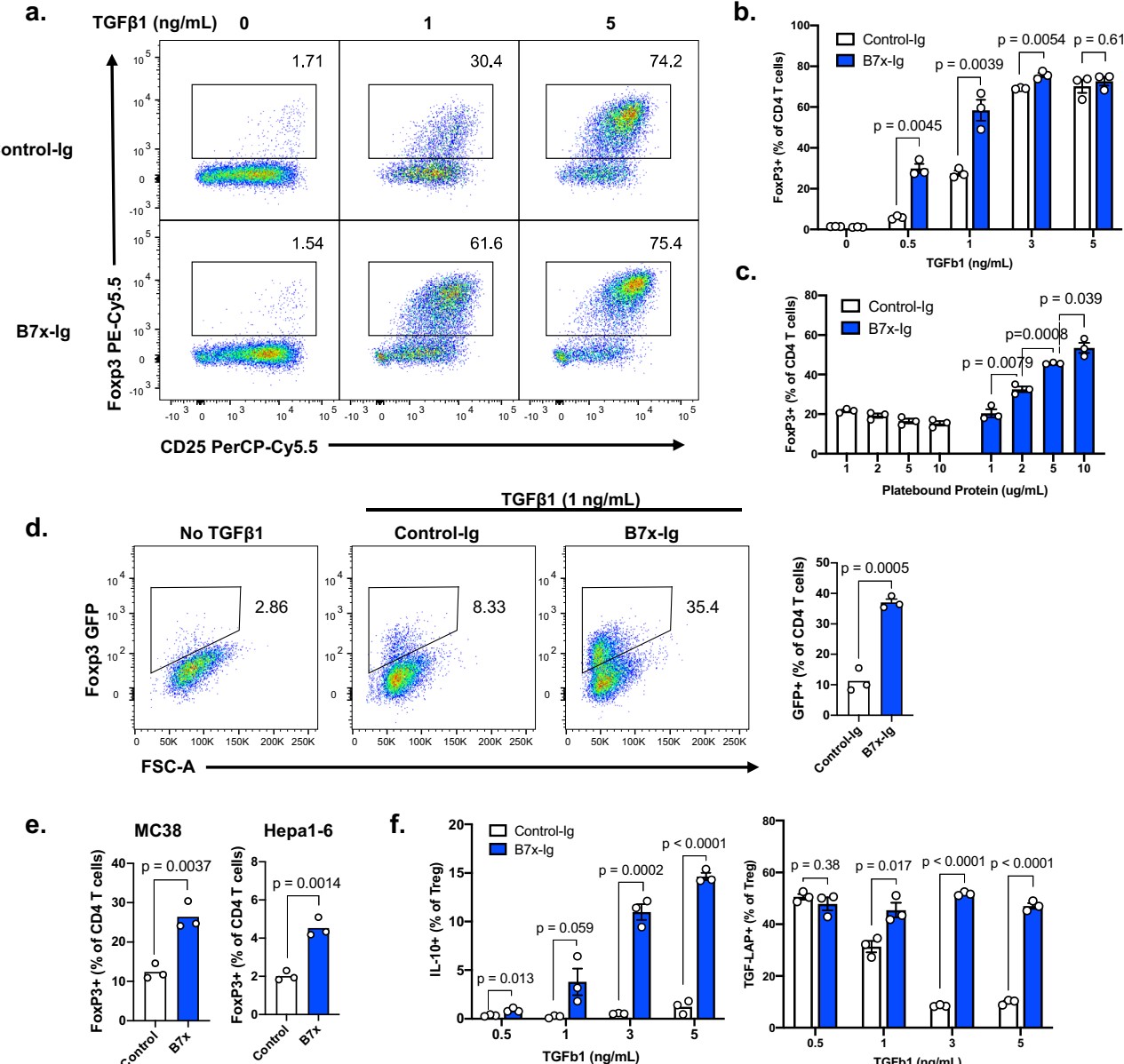

**Fig. 2 B7x promotes the expression of Foxp3 in CD4⁺ T cells. a, b** Splenic CD4⁺ T cells were isolated from wildtype mice and were cultured with iTreg-inducing conditions: anti-CD3, anti-CD28, IL-2, 10μg/mL recombinant B7x-Ig fusion protein or Control-Ig, and varying concentrations of TGFβ1, and expression of Foxp3 was measured by flow cytometry after 4 days. **c** CD4⁺ T cells were cultured as described in (**a**), except with fixed 1 ng/mL TGFβ1 and varying concentrations of B7x-Ig or Control-Ig, and expression of Foxp3 was assayed after 4 days. **d** CD4⁺ T cells from Foxp3-GFP/DTR mice were cultured in iTreg-inducing conditions with either B7x-Ig or Control-Ig and 2 ng/mL TGFβ1, and expression of GFP was measured after 4 days. **e** CD4⁺ T cells were cultured with anti-CD3, anti-CD28, and either B7x-transduced or control MC38 or Hepa1-6 cells, and expression of Foxp3 in the T cells was analyzed after 4 days. **f** iTregs were induced with B7x-Ig or Control-Ig as described in (**a**), and expression of intracellular IL-10 and surface TGF-LAP was measured after 4 days. Error bars represent SEM, P values were calculated by two-tailed Student's T-test. Results are representative of three independent experiments.

increased Treg: CD8 ratios (Supplementary Fig. 1c, f). In all tumor models studied, the infiltrating Tregs were similarly Nrp1⁻Helios⁺ (Supplementary Fig. 1d, g).

Next, we explored the correlation between B7x and Tregs in human cancers in silico. Analysis of RNA-seq data sets revealed positive correlations between *Vtcn1*, the gene name of B7x, and Foxp3 in multiple human cancers, as well as a highly statistically significant correlation in a pan-cancer analysis of 30 cancer types curated by The Cancer Genome Atlas (TCGA) (Supplementary Fig. 1h). Further, we analyzed the correlation between *Vtcn1* expression and the Treg gene signature using quanTIseq, a deconvolution algorithm that quantifies immune cell populations

in TCGA bulk RNA-seq data[28], and observed that *Vtcn1* positively correlates with the Treg gene signature in a wide range of cancers (Supplementary Fig. 1i).

Put together, these findings demonstrate that expression of B7x within tumors is associated with increased Treg populations in human cancers, and that B7x directly increases tumor-associated Treg populations in murine models.

**B7x induces Foxp3 expression in CD4⁺ T cells and enhances their immunosuppressive capacity.** Since B7x⁺ tumors contained greater numbers of Tregs, we hypothesized that B7x

directly acts upon conventional CD4$^+$ T cells and converts them to Tregs by inducing Foxp3 expression. To test this, we cultured splenic CD4$^+$ T cells in Treg-inducing conditions: anti-CD3/anti-CD28 stimulation, IL-2, varying concentrations of TGFβ1, and either recombinant fusion protein of B7x-Immunoglobulin-Fc (B7x-Ig) or isotype-control Fc protein (Control-Ig). This in vitro differentiation procedure generates Foxp3$^+$ induced Tregs (iTregs), thus mimicking the peripheral conversion of conventional CD4$^+$ T cells into Tregs. Consistent with our hypothesis, greater frequencies of T cells cultured with B7x-Ig expressed Foxp3 as compared to Control-Ig when TGFβ1 was present, although no effect was observed without TGFβ1, nor at very high levels of TGFβ1 where Foxp3 expression became saturated (Fig. 2a, b). When TGFβ1 concentrations were kept constant and the concentration of recombinant protein was titrated, B7x-Ig increased Foxp3 expression in a dose-dependent manner (Fig. 2c). We sought to determine whether B7x regulates Foxp3 at the transcriptional level, so we tested the capacity of B7x to drive the expression of GFP in CD4$^+$ T cells isolated from Foxp3-GFP/DTR mice, where eGFP is inserted downstream of Foxp3 at the native Foxp3 locus. We observed that B7x promoted the expression of eGFP in these cells, indicating that B7x enhances transcription of the Foxp3 gene (Fig. 2d). Further, CD4$^+$ T cells cultured in the presence of B7x$^+$ tumor cells expressed Foxp3 at greater rates than those cultured with B7x$^-$ tumor cells, consistent with our in vivo findings (Fig. 2e), and this effect was diminished by the addition of anti-TGFβ1 blocking antibody into the culture (Supplementary Fig. 3a). Thus, these results demonstrate that B7x directly binds to CD4$^+$ T cells and promotes the TGFβ1-dependent expression of Foxp3.

The expression of Foxp3 in CD4$^+$ T cells is associated with the acquisition of immunosuppressive properties, so we asked whether B7x enhances the capacity of iTregs to produce immunosuppressive cytokines. Consistent with our hypothesis, B7x-Ig induced greater expression of cell-surface TGF-LAP and intracellular IL-10 in Tregs (Fig. 2f), suggesting that B7x not only induces Foxp3 expression in CD4$^+$ T cells but makes them more functionally active.

Lastly, we revisited the possibility that B7x may expand nTreg populations by increasing their proliferation. We stimulated CTV-labeled GFP$^+$ CD4$^+$ T cells from Foxp3-GFP/DTR mice with anti-CD3/CD28 beads and either B7x-Ig or Control-Ig, then measured their proliferation by CTV dilution. B7x reduced the proliferation of Tregs, which could only be rescued by addition of exogenous IL-2 (Supplementary Fig. 3b). This is consistent with our in vivo finding of reduced Ki67 levels in tumor-infiltrating Tregs from B7x$^+$ tumors (Fig. 1f). Altogether, we find that B7x does not increase the proliferation of Tregs but instead promotes the conversion of conventional CD4$^+$ T cells into Tregs in a TGFβ1-dependent manner.

**B7x promotes a suppressive and activated Treg phenotype.** To obtain a comprehensive view of how B7x regulates the Treg phenotype, we employed whole transcriptome RNA sequencing (RNA-seq) in a two-pronged approach. First, we flow-sorted GFP$^+$ tumor-infiltrating Tregs from MC38-B7x and MC38-Control tumors engrafted in Foxp3-GFP/DTR mice and performed RNA-seq on the sorted Tregs. Second, we differentiated splenic CD4$^+$ T cells from Foxp3-GFP/DTR mice into iTregs in vitro in the presence of either B7x-Ig or Control-Ig, and subsequently flow-sorted GFP$^+$ iTregs for RNA-seq. Thus, we were able to analyze the effects of B7x on Tregs in the in vivo tumor context, as well as the immediate transcriptomic effects of B7x signaling during Treg differentiation in vitro.

When comparing tumor-infiltrating Tregs, we observed over 7000 differentially expressed genes (DEGs) between Tregs isolated from MC38-B7x vs. MC38-Control tumors (Fig. 3a, b). Indeed, Gene Ontology analysis indicated global transcriptomic changes in pathways associated with protein translation, metabolism, and cell division (Fig. 3c). Focusing on genes associated with immunosuppressive function, we noted that Treg effector genes such as *Pdcd1, Prf1, Gzmb, Il10, Entpd1, Itgb8,* and *Tigit* were upregulated in B7x-associated Tregs (Fig. 3b). For an unbiased analysis of the Treg functional state, we employed Gene Set Enrichment Analysis, and found that Tregs isolated from MC38-B7x tumors were enriched in gene sets associated with Treg suppression and effector functions[29] (Fig. 3d). Overall, Tregs in B7x-expressing tumors possessed a more activated and suppressive transcriptome.

Next, we explored how B7x affects the immediate intracellular signaling pathways using our RNA-seq analyses of in vitro differentiated iTregs (Fig. 3e, f). We observed broad alterations in genes associated with T cell identity and differentiation, including *Tcf7, Stat1, Fos, Jun, Eomes, Ezh2,* and *Foxo1* (Fig. 3g). When combining the in vivo and in vitro data, we observe over 470 DEGs common to both data sets, and a consistent upregulation of Treg-associated effector genes such as *Pdcd1, Entpd1,* and *Itgae* (Fig. 3h).

To confirm that the Treg transcriptomic changes resulted in increased suppressive capacity, we performed a proliferation suppression assay: Control and B7x-induced iTregs were co-cultured with CellTrace Violet-labeled responder CD4$^+$ or CD8$^+$ T cells, and the proliferation of the responder T cells was analyzed. B7x-induced iTregs demonstrated significantly greater suppression of both CD4$^+$ (Fig. 3i) and CD8$^+$ T cell (Fig. 3j) proliferation, indicating that B7x not only promotes Foxp3 expression in CD4$^+$ T cells, but also confers greater suppressor function.

Put together, B7x signaling in Tregs promotes a phenotype consistent with increased activation and immunosuppression.

**B7x regulates iTreg differentiation via the Akt-Foxo1 pathway.** Little is known about the intracellular signal transduction pathway initiated by B7x in CD4$^+$ T cells. We sought to elucidate the signaling pathways that underly B7x-mediated Treg induction via phospho-flow cytometry. Since B7x is an immune checkpoint with known inhibitory effects on T cell activation, we first analyzed the phosphorylation status of transcription factors downstream of the T cell receptor (TCR) complex in CD4$^+$ T cells cultured in Treg-inducing conditions with either B7x-Ig or Control-Ig. We observed that cells cultured with B7x exhibited decreased phosphorylation of the NF-κB subunit p65 and the AP-1 subunit c-Jun, confirming TCR signaling is decreased by B7x (Fig. 4a, b).

To understand how B7x could be regulating the iTreg differentiation process, we next screened transcription factors of the cytokine signaling pathways that are relevant to CD4$^+$ T cell polarization. In each of these factors, phosphorylation indicates increased activation and transcriptional activity. The TGFβ pathway is a key activator of Foxp3 expression[30], but interestingly, we noted decreased phosphorylation of the downstream transcription factor complex Smad2/3 in B7x-induced iTregs, indicating this pathway is unlikely to explain why Foxp3 expression is increased in B7x-induced cells (Supplementary Fig. 4a). However, B7x also inhibited STAT3 phosphorylation, which plays important roles in differentiation of CD4$^+$ T cells into the T$_H$17 subtype[31] (Supplementary Fig. 4c). B7x did not have any observable effect on STAT1 or STAT4 phosphorylation (Supplementary Fig. 4b, d). The IL-2 pathway is essential to iTreg differentiation and acts through STAT5[30], and we observed that B7x enhances STAT5 phosphorylation, which may contribute to

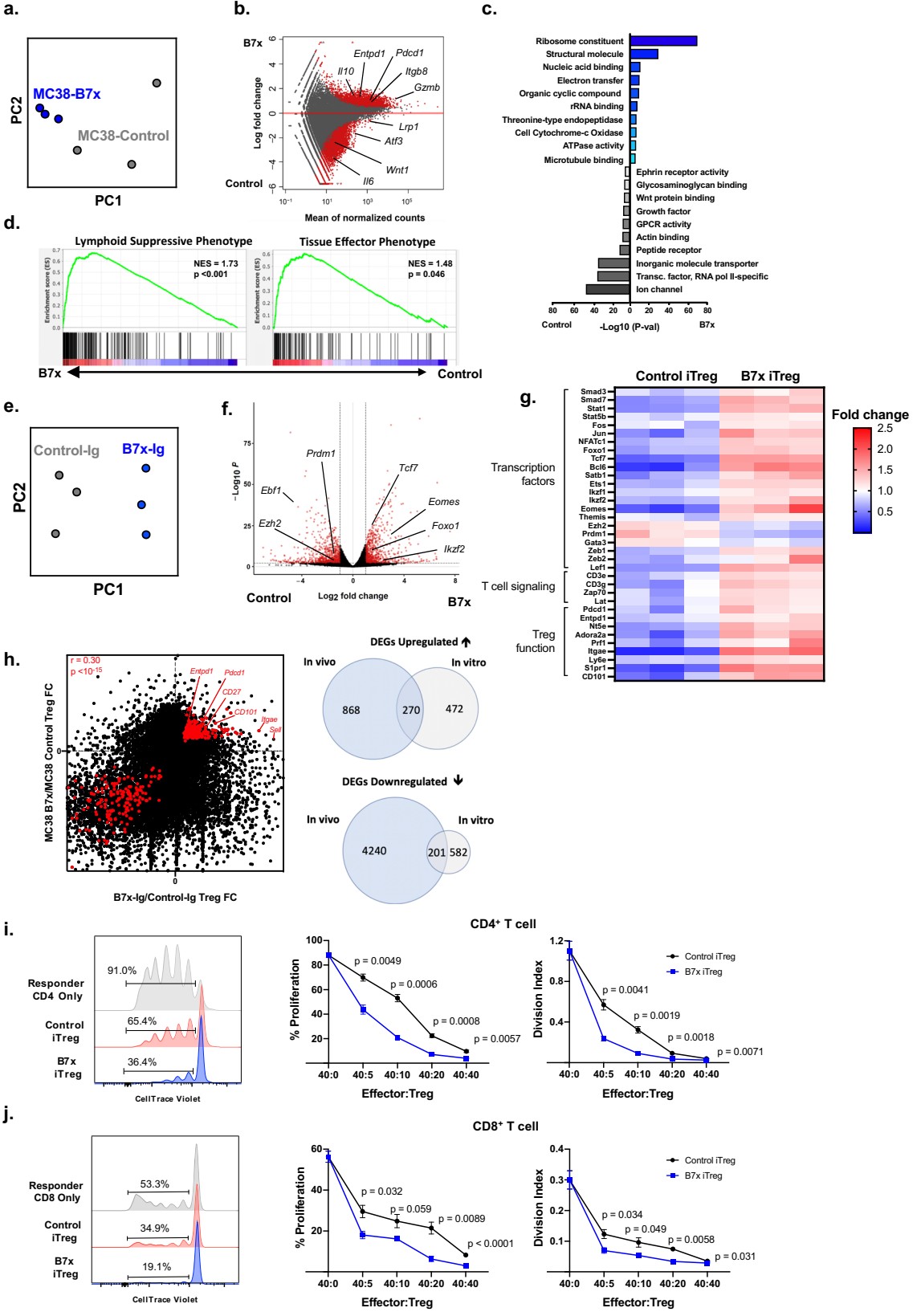

B7x-mediated Foxp3 expression (Supplementary Fig. 4e). Moreover, we analyzed the levels of mitochondrial reactive oxygen species (mitoROS) which has recently been implicated to promote iTreg induction[32], but found that B7x does not promote and instead reduces mitoROS in iTregs (Supplementary Fig. 4f).

We next examined the critical pathway mediated by the serine-threonine kinase Akt, which lies downstream of the TCR and has an essential role in $CD4^+$ T cell differentiation[33]. We observed that phosphorylation of Akt was decreased in iTregs cultured with B7x, indicating decreased Akt activity which is consistent

**Fig. 3 B7x promotes immunosuppressive function in Tregs. a–d** MC38-B7x and MC38-Control tumors were engrafted into Foxp3-GFP/DTR mice, and after 7 days, tumors were dissociated and GFP$^+$ CD4$^+$ T cells were flow-sorted for RNA extraction and whole transcriptome RNA-seq analysis, $n = 3$ samples per group. **a** Principle component analysis (PCA) plots based on all expressed genes in MC38-B7x and MC38-Control tumors. **b** Microarray (MA) plot showing gene expression changes between the two Treg groups, red dots indicate significantly differentially expressed genes based on log$_2$fold change ≥1 or ≤−1 and adjusted $p$-value < 0.05 as calculated by the Benjamini and Hochberg method. **c** Gene ontology (GO) analysis on molecular functions is shown, with GO terms enriched in Tregs from B7x$^+$ tumors on the right and Tregs from B7x$^-$ control tumors on left. **d** Gene set enrichment analysis was performed on DEGs with gene sets for Treg Suppressive and Effector phenotypes, normalized enrichment score (NES) and familywise error rate (FWER) $p$ values based on modified Kolmogorov Smirnov test are shown in top-right of each plot. **e–g** CD4$^+$ T cells from Foxp3-GFP/DTR mice were induced into iTregs in vitro in the presence of either Control-Ig or B7x-Ig, and subsequently GFP$^+$ CD4$^+$ T cells were flow-sorted for RNA extraction and RNA-seq ($n = 3$ samples per group). **e** PCA plots based on all expressed genes in the Control-Ig and B7x-Ig Tregs. **f** Volcano plot of differential gene expression comparing Control-Ig Tregs to B7x-Ig Tregs, genes in red are upregulated in either group at log$_2$ fold change ≥1 or ≤−1 and adjusted $p$-value < 0.05 as calculated by the Benjamini and Hochberg method. **g** Double-gradient heatmap depicting fold change of selected DEGs relevant to Treg differentiation for each sample in the Control-Ig and B7x-Ig Treg groups. **h** All genes from the datasets of (**b**) and (**f**) were plotted by magnitude of fold change, genes appearing as DEGs in both data sets are displayed in red (left). Venn diagrams listing numbers of DEGs in each data set are shown (right). **i, j** iTregs were generated with B7x-Ig or Control-Ig as described in (**d**), and resulting GFP$^+$ iTregs were flow-sorted and co-cultured with CellTrace Violet (CTV)-labeled responder T cells and anti-CD3/CD28 Dynabeads, and proliferation of responder T cells was analyzed by CTV dilution after 3 days. Error bars represent SEM, $P$ values were calculated by two-tailed Student's $T$-test. Results are representative of three independent experiments.

with our prior observation of decreased TCR signaling in B7x-treated cells (Fig. 4c). Looking downstream of Akt, we asked whether Akt regulates any transcription factors that modulate Foxp3 expression. Akt has been described to phosphorylate and inhibit the Foxo-family factors such as Foxo1, which serve as important activators of Foxp3 expression in iTregs[34]. Upon phosphorylation by Akt, Foxo1 translocates out of the nucleus and into the cytoplasm, where it is inactive. In iTregs cultured with B7x-Ig, we observed similar total cellular levels of Foxo1 (Fig. 4d), but there was a marked decrease in Foxo1 phosphorylation (Fig. 4e). Since Foxo1 is active in its dephosphorylated form, we calculated the proportion of Foxo1$^+$ cells that were p-Foxo1$^-$ and found that B7x-treated iTregs had markedly increased levels of active Foxo1 (Fig. 4f). We further confirmed that B7x reduces Akt and Foxo1 dephosphorylation by western blot (Supplementary Fig. 4g). Next, we analyzed the cellular localization of Foxo1 by confocal microscopy and confirmed that B7x-treated iTregs had greater localization of Foxo1 in the cell nucleus (Fig. 4g). B7x-treated iTregs had greater quantitative correlation of Foxo1 with the nuclear marker DAPI (Fig. 4h) and a greater percentage of B7x-treated cells exhibited a nuclear-localized phenotype of Foxo1 (Fig. 4g, i).

We asked whether the B7x-mediated effects on Akt and Foxo1 phosphorylation are also seen in nTregs. We found that B7x inhibited Akt phosphorylation in nTregs, but this could be rescued by addition of IL-2 (Supplementary Fig. 3c), consistent with our earlier findings that nTreg proliferation was inhibited by B7x but was relieved with addition of IL-2 (Supplementary Fig. 3b). Interestingly, B7x did not affect phosphorylation of Foxo1 in nTregs (Supplementary Fig. 3d), potentially indicating that the B7x-Foxo1 pathway is specific for iTregs.

To test whether the Foxo-pathway is indeed necessary for B7x-mediated enhancement of iTregs, we used chemical inhibitors targeting the Akt/Foxo pathway to determine how this affects Foxp3 expression in iTregs generated with B7x or Control-Ig. In Control iTregs, we observed that Akt inhibition enhanced iTreg differentiation, consistent with the known role of Akt in hindering iTreg development (Fig. 4j). In contrast, Akt inhibition yielded no greater Foxp3 expression in B7x-treated cells, indicating that maximal Akt inhibition had already been achieved by the B7x pathway. In contrast, inhibition of the Foxo pathway drastically reduced the B7x-mediated increase in Foxp3 expression, confirming the critical role the Foxo transcription factors plays in iTreg induction. We next investigated which phosphatases may be responsible for dephosphorylating Akt. Whereas inhibition of the phosphatase Shp1 reduced B7x-mediated

enhancement of Foxp3, inhibition of Shp2 or PTEN did not significantly reduce the effect of B7x (Fig. 4j, Supplementary Fig. 4h). Lastly, we examined mTOR which lies downstream of Akt to determine whether B7x-mediated inhibition of Akt also suppressed mTOR signaling. Inhibition of mTOR with rapamycin promoted Foxp3 expression in both control and B7x-treated iTregs, suggesting that the effect of B7x-mediated inhibition of Akt is largely restricted to the Foxo pathway and not mTOR pathway (Supplementary Fig. 4h).

Collectively, our data demonstrate that B7x inhibits TCR signaling in Treg-inducing conditions which subsequently suppresses Akt and activates Foxo1 to promote Foxp3 expression.

**Tumor-expressed B7x mediates Treg-dependent resistance to anti-CTLA-4.** Since B7x broadly alters the Treg phenotype, we asked whether B7x modified the expression of Treg proteins that are targets for immunotherapy. In particular, we analyzed the expression of the cell-surface receptor CTLA-4, a protein that is typically expressed by Tregs and is the target for immune checkpoint blockade. Moreover, anti-CTLA-4 antibody therapy has been shown to cause Treg depletion in murine and human tumors by Fc-mediated depletion of antibody-bound cells[35–38]. Interestingly, we found that B7x reduced the expression of cell-surface CTLA-4 on tumor-infiltrating Tregs in vivo (Fig. 5a) and induced Tregs in vitro (Fig. 5b). Therefore, we hypothesized that the B7x-mediated reduction of CTLA-4 would make tumor-infiltrating Tregs poorer targets for immunotherapy and thus reduce the efficacy of anti-CTLA-4 therapy.

To test this, we engrafted mice with MC38-B7x and MC38-Control tumors and then treated the mice with either anti-CTLA-4 or IgG isotype control antibody (Fig. 5c). Whereas anti-CTLA-4 treatment successfully reduced tumor growth in MC38-Control tumors, it did not significantly reduce the growth of MC38-B7x tumors, demonstrating a phenotype of resistance to anti-CTLA-4 treatment in B7x-expressing tumors (Fig. 5d, e). After 17 days, we dissociated the tumors to analyze the immune infiltrate (Supplementary Fig. 5a–c). At this late time-point, little difference was observed in Treg populations between MC38-B7x and MC38-Control tumors of untreated mice; by this point, tumors in both groups had grown large and the Treg-promoting effect of B7x was lost (Fig. 5f–h). In contrast, a marked difference was observed on Treg populations between the two tumor types when the mice were treated with anti-CTLA-4. Anti-CTLA-4 treatment was effective in depleting Treg populations in MC38-Control tumors, however, it

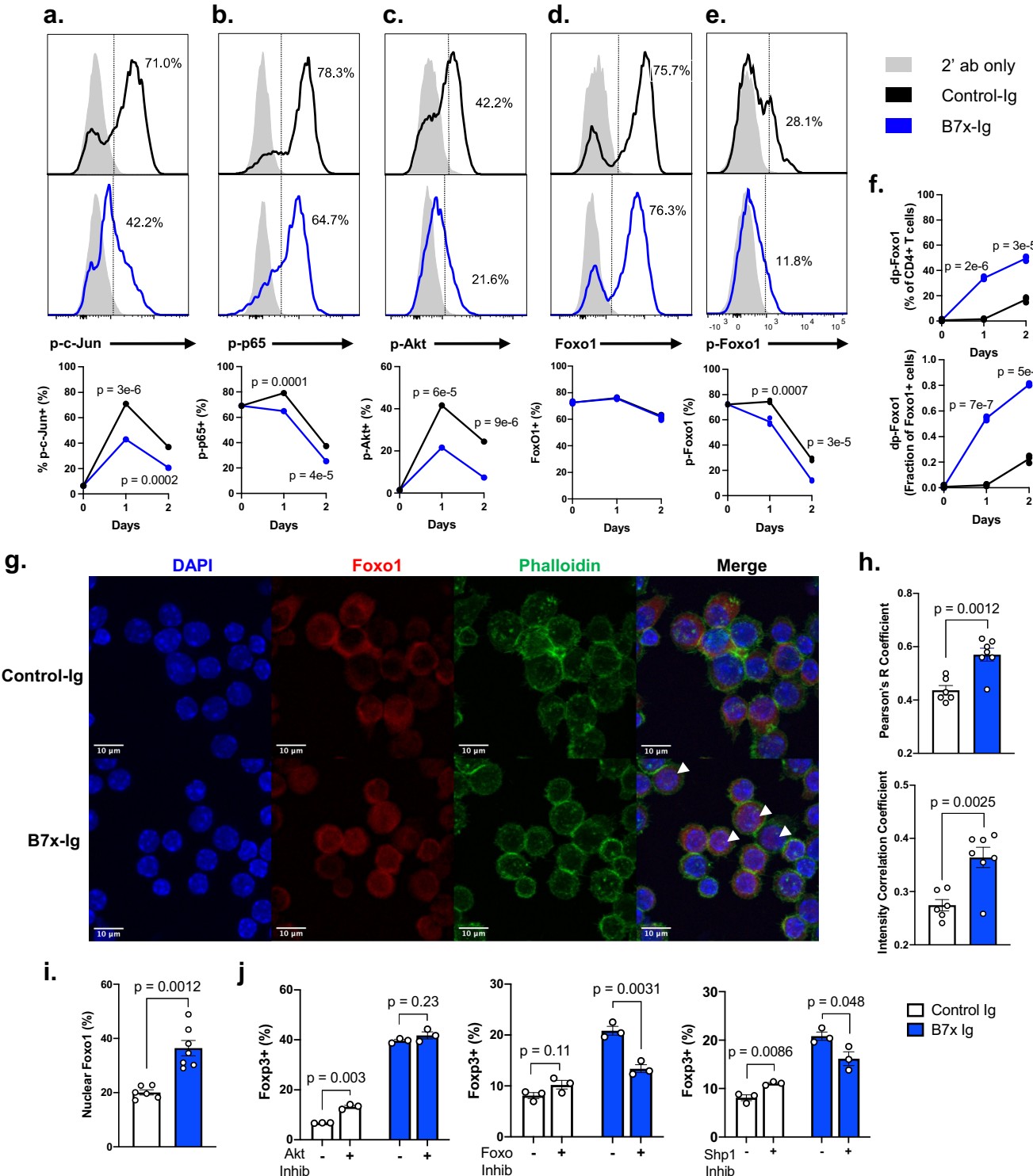

proved to be ineffective in reducing Treg populations in MC38-B7x tumors (Fig. 5f–h). We broadly profiled the tumor immune infiltrate and found that anti-CTLA-4 treatment generally increased effector T cell and NK cell populations in B7x[+] and B7x[−] tumors (Fig. 5f, g). In the myeloid compartment, we observed that anti-CTLA-4 treatment caused a shift of F4/80[+] macrophages to F4/80[−] Ly6C[+] monocytes in both tumor types (Fig. 5f, g). The effects of anti-CTLA-4 on cell populations was similar between MC38-B7x and MC38-Control tumors, with the notable exception of Tregs, which were not effectively depleted in MC38-B7x tumors (Fig. 5h).

Next, we asked whether the differences in Treg populations would affect the activation status of effector immune cells. Anti-CTLA-4 treatment greatly increased IFN-γ production in CD4[+] T cells, CD8[+] T cells, natural killer (NK) cells, and NKT cells in MC38-Control tumors, but the anti-CTLA-4-mediated increase in IFN-γ was markedly less in MC38-B7x tumors (Fig. 5i). This indicates reduced functional capacity of effector T and NK cell populations in B7x[+] tumors after anti-CTLA-4 treatment, which reflects the increased Treg populations in these tumors.

To determine whether Tregs were necessary for the B7x-mediated resistance to anti-CTLA-4, we eliminated Tregs from

**Fig. 4 B7x regulates the Akt-Foxo1 pathway to enhance Foxp3 expression. a–f** CD4$^+$ T cells were cultured in iTreg-inducing conditions with either B7x-Ig or Control-Ig for up to 48 h, and phosphorylation status of c-Jun (**a**), p65, **b**, Akt, **c**, total Foxo1 (**d**), and phospho-Foxo1 (**e**) was determined by phospho-flow cytometry. **f** The percentage of cells with dephosphorylated Foxo1 (% total Foxo1$^+$ – % p-Foxo1$^+$) and fraction of dephosphorylated Foxo1 of Foxo1$^+$ cells are shown. Error bars represent SEM, P values were calculated by two-tailed Student's T-test, unadjusted for multiple comparison. Results are representative of 4 independent experiments. **g–i** iTregs were generated as described in (**a**) and were subsequently stained with nuclear marker DAPI, cytoplasmic marker phalloidin, and anti-Foxo1, and were analyzed by confocal microscopy. **g** Representative images are shown, arrowheads represent nuclear localization of Foxo1. **h** Colocalization analysis for overlap of Foxo1 and DAPI signal was performed to calculate Pearson's R coefficient and Intensity Correlation Coefficient, each point represents a unique field-of-view from separate experiments ($n = 6$ per group). Error bars represent SEM. Representative of three independent experiments. **i** Percent of cells with nuclear localization of Foxo1 per field-of-view. **j** CD4$^+$ T cells were induced into iTregs with either B7x-Ig or Control-Ig and were treated with inhibitors against Akt (GSK690693), Foxo1/3 (AS1842856, carbenoxolone), and Shp1 (TPI-1), and Foxp3 expression was analyzed after 4 days. Error bars represent SEM, P values were calculated by two-tailed Student's T-test. Results are representative of three experiments.

tumors in vivo using the diphtheria toxin receptor (DTR) depletion model. Foxp3-GFP/DTR mice were engrafted with MC38-B7x or MC38-Control tumors and subsequently treated with human diptheria toxin (DT), which eliminated tumor-infiltrating Tregs within 2 days (Supplementary Fig. 6a, b). MC38-B7x tumors were not able to survive anti-CTLA-4 treatment in DT-treated mice. Indeed, DT-mediated Treg depletion was highly effective in reducing tumor growth in both MC38-Control and MC38-B7x tumors, demonstrating the critical importance that Tregs have in tumor survival and growth (Supplementary Fig. 6c). Next, because Tregs represent a cell-extrinsic mode of resistance to treatment, we hypothesized that a mixed MC38-Control/B7x tumor would similarly be resistant to treatment as the MC38-B7x tumors. Therefore, we mixed MC38-B7x and MC38-Control cells at a 1:1 ratio, engrafted this mixed tumor into mice, and then treated mice with either anti-CTLA-4 or isotype antibody. The mixed MC38 tumor exhibited resistance to treatment similar to MC38-B7x tumors (Supplementary Fig. 6d, e). Upon dissociation of the mixed tumors, we observed that both B7x$^+$ and B7x$^-$ tumor cells were present in equal proportions, indicating the survival benefit provided by B7x extended to both cell populations (Supplementary Fig. 6f, g). Collectively, these findings show that tumor-expressed B7x provides resistance to anti-CTLA-4 therapy in Treg-dependent manner.

Given the findings thus far, we suspected that B7x-mediated resistance to anti-CTLA-4 therapy was largely due to the effects of B7x directly on the Treg population. However, we also considered that B7x may hinder the capacity of nearby myeloid cells to perform antibody-mediated depletion of Tregs. The Treg-depleting effect of anti-CTLA-4 has been shown to be Fc-mediated, primarily by tumor-infiltrating macrophages (TAMs) expressing FcγRI and FcγRIV within the tumor microenvironment[35]. Therefore, we explored whether B7x reduced the capacity of these cells to perform antibody-dependent depletion of Tregs. First, we measured the surface expression of FcγRI and FcγRIV in F4/80$^+$ tumor-infiltrating macrophages in MC38-Control and MC38-B7x tumors, and found no appreciable differences (Supplementary Fig. 6g). Next, we assayed the functional capacity of these TAMs by flow-sorting CD11b$^+$ F4/80$^+$ cells from MC38-Control or MC38-B7x tumors and co-culturing them with FITC-labeled phagocytosis beads, and again, found no difference in the bead uptake from either macrophage group (Supplementary Fig 6h). Moreover, since CTLA-4 binds the ligands CD80 and CD86, we analyzed the expression of these ligands on both TAMs and the tumor cells and found no difference in the expression of these molecules on either macrophages or tumor cells (Supplementary Fig 6i, j). Thus, we found no differences in the functional capacity of the tumor-infiltrating macrophage populations, indicating that reduced functional status in these cells is unlikely to explain why Tregs are not effectively depleted by anti-CTLA-4 treatment in B7x-expressing tumors.

Altogether, our data suggest that due to B7x-mediated alteration of the CTLA-4 expression on Tregs and the expansion of Tregs, anti-CTLA-4 therapy is less effective in depleting Treg populations and therefore has reduced therapeutic efficacy in B7x$^+$ tumors.

**Anti-B7x and anti-CTLA-4 combination treatment has synergistic therapeutic efficacy.** Since tumor-expressed B7x reduced the efficacy of anti-CTLA-4 treatment, we next asked if this phenotype could be overcome by administering an anti-B7x antibody. We used our anti-B7x antibody clone 1H3 which we previously demonstrated to have therapeutic effect against B7x$^+$ tumor metastases to the lung[6]. Against MC38 cells, clone 1H3 specifically binds to B7x$^+$ tumor cells but not B7x$^-$ cells (Fig. 6a). So, we engrafted mice with MC38-B7x tumors and administered anti-CTLA-4, anti-B7x, combination of anti-B7x and anti-CTLA-4, or IgG isotype control (Fig. 6b). Anti-CTLA-4 monotherapy had no observable effect as shown in Fig. 5, whereas anti-B7x monotherapy had limited but not durable response. However, the combination of both had synergistic efficacy and significantly reduced tumor growth (Fig. 6c).

When examining the tumor infiltrate, combination anti-B7x and anti-CTLA-4 treatment caused broad changes in the tumor microenvironment (Fig. 6d). In the T cell population, effector T cell populations were markedly increased with combination treatment. Total CD4$^+$ T cell infiltration was increased with combination treatment (Fig. 6e), with a specific increase in Foxp3$^-$ effector CD4$^+$ T cells, and a relative decrease in Tregs as a proportion of CD4$^+$ T cells (Fig. 6f). Simultaneously, a greater shift of macrophages to monocytes was observed with combination treatment (Fig. 6d, e). These changes in cell populations also correlated with increased IFN-γ production in CD4$^+$ T cells, CD8$^+$ T cells, and NK cells, indicating increased effector functions (Fig. 6g).

In sum, B7x promotes Treg differentiation and thus mediates resistance to anti-CTLA-4 treatment, but B7x-mediated resistance can be overcome by the use of anti-B7x antibody. These results show that B7x has multiple immunosuppressive roles in the tumor microenvironment and is a promising target for combination immunotherapy.

**Discussion**

Compared to other B7-family members like PD-L1, B7x is much less studied, and its mechanisms of action are not well understood. B7x is often associated with advanced disease progression and poorer clinical outcomes in human cancers, and in this work, we describe a mechanism by which it acts in the tumor microenvironment. Here, we show that tumor-expressed B7x expands Treg populations in mouse and human cancers (Fig. 1). B7x promotes Treg populations by inducing Foxp3 expression in conventional CD4$^+$ T cells and converting them into Tregs in a TGFβ1-dependent manner (Fig. 2). B7x-mediated induction of

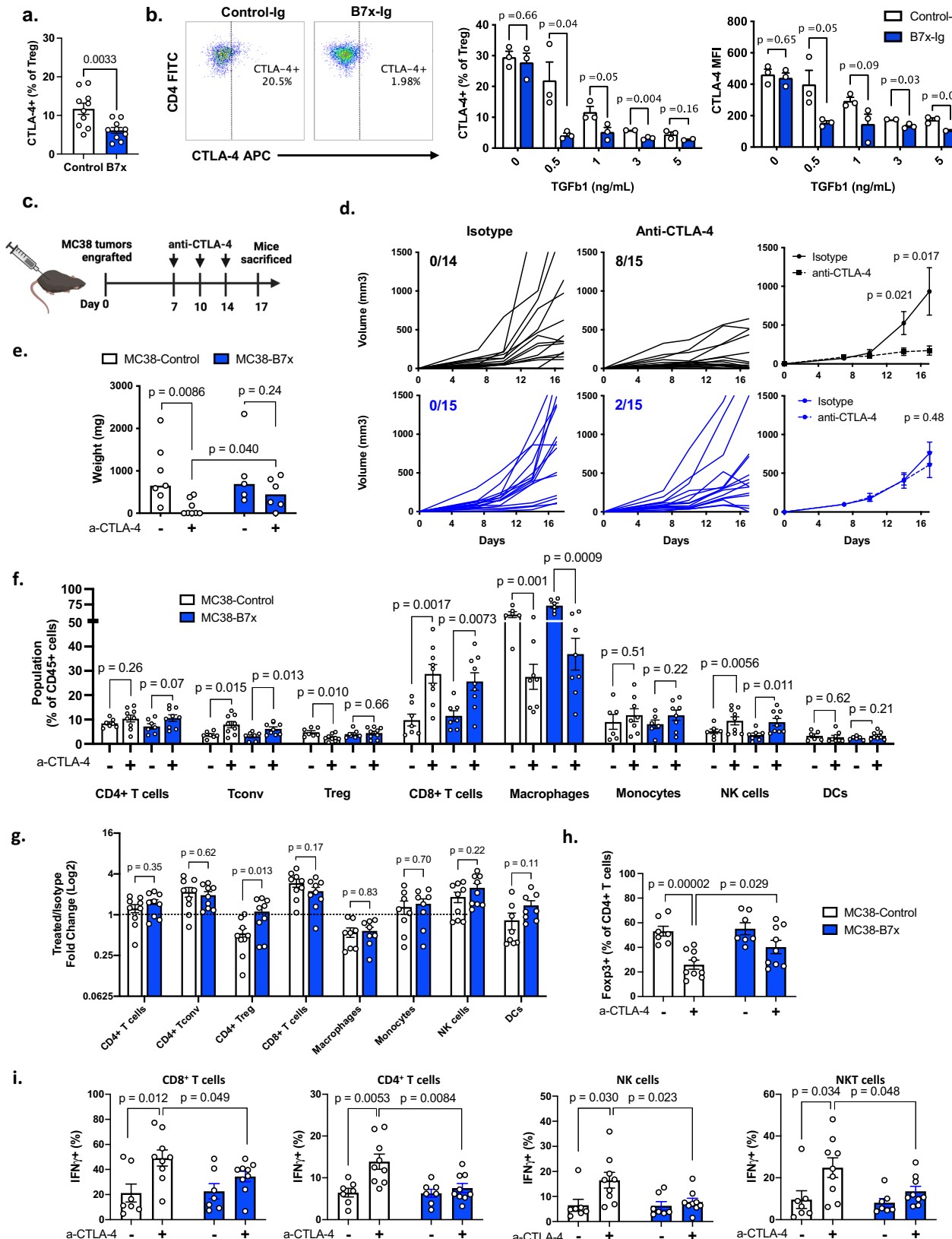

Foxp3 is concurrent with increased immunosuppressive functions in vivo and in vitro (Fig. 3). B7x inhibits Akt and activates Foxo signaling in CD4[+] T cells, which is necessary for B7x-mediated increase of Foxp3 expression (Fig. 4). Expression of B7x in tumors reduces the efficacy of anti-CTLA-4 therapy by impairing

Treg depletion (Fig. 5), but this phenotype of resistance can be overcome by combination of anti-B7x and anti-CTLA-4 treatment (Fig. 6).

Our observations that B7x can drive Treg induction from conventional CD4[+] T cells demonstrate that it has additional

**Fig. 5 Expression of B7x in tumors reduces efficacy of anti-CTLA-4 treatment. a** Mice were engrafted with MC38-Control or MC38-B7x tumors, sacrificed after 7 days, and tumors were dissociated to analyze expression of CTLA-4 on Foxp3$^+$ Tregs. $n = 10$ mice per group, representative of three independent experiments. **b** Splenic CD4$^+$ T cells were differentiated into Tregs with either Control-Ig or B7x-Ig and varying amounts of TGFβ1, and expression of CTLA-4 was analyzed after 4 days. Results are representative of 3 independent experiments. Representative flow cytometry plots (left), and graphs quantifying percent CTLA-4$^+$ of total Tregs and CTLA-4 MFI of Tregs are shown (right). **c** Experimental scheme of MC38 tumor engraftment and anti-CTLA-4 (+) or IgG isotype (−) treatment. **d** Tumor volumes were tracked, individual tumor volumes (left) were tracked and numbers designate tumor clearance (final tumor volume <50 mm$^3$), and mean tumor volumes were calculated (right). There were 14 mice in the MC38-Control isotype-treated group, 15 mice in the MC38-Control anti-CTLA-4-treated group, 15 mice in the MC38-B7x isotype-treated group, and 15 mice in the MC38-B7x anti-CTLA-4 treated group. Representative of five independent experiments. **e** Mice were engrafted with tumors and treated as described in (**c**), sacrificed at day 17, and tumors were extracted and weighed. There were seven mice in the MC38-Control isotype-treated group, eight mice in the MC38-Control anti-CTLA-4-treated group, five mice in the MC38-B7x isotype-treated group, and six mice in the MC38-B7x anti-CTLA-4 treated group. Representative of three independent experiments. **f, g** Mice were engrafted with tumors and treated as in (**c**), mice were sacrificed for tumor dissociation at day 17, and cells were analyzed by flow cytometry. CD45$^+$ immune cells were quantified and calculated as proportion of total immune cell population (**f**), and fold change of the anti-CTLA-4 treated groups normalized to each group's Isotype-treated population (**g**). **h** Proportions of Foxp3$^+$ cells of CD4$^+$ T cells were determined. **i** Cell suspensions were stimulated with PMA/ionomycin and were stained intracellularly for IFN-γ. For (**f–i**), there were seven mice in the MC38-Control isotype-treated group, nine mice in the MC38-Control anti-CTLA-4-treated group, seven mice in the MC38-B7x isotype-treated group, and 9 mice in the MC38-B7x anti-CTLA-4 treated group. Representative of three independent experiments. Error bars represent SEM, P values were calculated by two-tailed Student's T-test.

roles in the tumor microenvironment in parallel to its established role as an inhibitory ligand to effector T cells. Immune checkpoints are best known for their role in modulating T cell activation by inhibiting TCR-mediated signaling, which has major ramifications on the priming, expansion, and exhaustion of CD8$^+$ T cells[39,40]. While many of these functions also apply to CD4$^+$ T cells, CD4$^+$ T cells exhibit great plasticity in terms of helper-type polarization, making the roles of immune checkpoints in these cells complex. Immune checkpoints play major roles in influencing CD4$^+$ T cell polarization. The B7-family coinhibitory ligand PD-L1 promotes iTreg differentiation by signaling through its receptor PD-1[41,42], whereas the costimulatory receptor CD28 has been shown to suppress iTreg differentiation[43]. Here, we show that B7x inhibits TCR signaling as a coinhibitory ligand and promotes Foxp3 expression by modulating the Akt/Foxo pathway. This points to a shared regulatory network governed by costimulatory and coinhibitory pathways that determines polarization of CD4$^+$ T cells to inflammatory or immunosuppressive subtypes. This may be especially significant in the context of the tumor immune landscape, where distinct patterns of immune checkpoint expression may influence the tumor-infiltrating CD4$^+$ T cell milieu.

In this work, we show that B7x signaling reduces the expression of CTLA-4 on the cell surface of Tregs, reducing the efficacy of anti-CTLA-4 against these cells. The expression and cell-surface localization of CTLA-4 in T cells is dynamically regulated. In conventional T cells, CTLA-4 is expressed at low levels in the resting state, and is trafficked to the cell-surface upon TCR-mediated activation, whereupon it is subsequently endocytosed and internalized to limit its activity[44]. In contrast, Tregs constitutively express high levels of CTLA-4, which can be further modulated with stimulation[45]. As B7x inhibits TCR-signaling in Tregs (Fig. 4a–c), it is not surprising that CTLA-4 expression is also affected. This can have major ramifications on the efficacy of agents that target CTLA-4, as we show here (Fig. 5b). Moreover, tumor-expressed B7x may also alter the expression of other receptors that are highly expressed on Tregs such as OX-40, GITR, and CCR4, which have also shown to have Treg-depleting effects when targeted by therapeutic antibodies[46]. Further work will be necessary to explore these possibilities.

Immune checkpoint blockade is a proven therapeutic strategy for many cancer types, however, resistance to therapy remains one of the most significant obstacles. When one checkpoint molecule is targeted for blockade, the expression of alternate immune checkpoints reduces efficacy of the therapy[8]. In this

study, we demonstrate that expression of B7x in tumors reduces the efficacy of anti-CTLA-4 therapy in a Treg-dependent manner. Interestingly, expression of B7x has been shown to reduce efficacy of anti-PD-1 therapy in non-small cell lung cancer[47]. However, since anti-PD-1 therapy does not exhibit Treg-depleting properties like anti-CTLA-4, the mechanism of resistance mediated by B7x to anti-PD-1 is likely to rely more heavily on its inhibitory effects on effector T cells. This suggests that the most important role of B7x within the tumor microenvironment is context-dependent.

Combination therapy is one of the major strategies to improve response rates to cancer immunotherapy. In particular, combined anti-PD-1 and anti-CTLA-4 treatment exhibits greater response rates than monotherapy of either agent in melanoma[48], lung cancer[49], and colon cancer[50], albeit at the cost of greater immune-related adverse effects[51]. Therefore, there is a great need for rational therapeutic combinations that maximize therapeutic benefit while minimizing toxicity. Considering the widespread expression of B7x in many cancer types, it is a sensible target for blockade, for which monotherapy has proven to be effective in murine models[6,10]. Here, we show that anti-B7x achieves synergistic therapeutic efficacy in combination with anti-CTLA-4 treatment, indicating that combining anti-B7x with currently available checkpoint inhibitors may be a promising strategy to improve clinical response against B7x$^+$ tumors. Further, anti-B7x has shown a favorable safety profile in early clinical trials[52], making it a promising candidate for combination therapy with potentially low toxicity. The efficacy and utility of anti-B7x in monotherapy or combination therapy will require future studies to verify.

## Methods

**Mice.** All animal studies were performed in accordance with protocols that were reviewed and approved by the Institutional Animal Care and Use Committee (IACUC) of the Albert Einstein College of Medicine. Wild type 8–12-week-old female C57BL/6 (B6) and BALB/c mice were purchased from Charles River (Frederick, MD). B6 Foxp3-GFP-DTR and B6 CD45.1 mouse strains were obtained from Jackson Laboratories (Bar Harbor, ME). All animals were housed in a Specific-Pathogen-Free facility, and were kept in-house for at least 2 weeks prior to use for experiments. Animals were housed under 12 light/12 dark cycles, maintained at 65–75 F at 30–60% humidity.

**T cell isolation.** CD4$^+$ and CD8$^+$ T cells were isolated from the spleens of C57BL/6 mice. Briefly, spleens were crushed and strained through a 100 μm cell strainer, and then resuspended in red blood cell lysis buffer. The cell suspension was strained again through a 40 μm cell strainer, and then magnetically sorted with the CD4 or CD8 T Lymphocyte Enrichment Kit (BD). After purification, T cells were

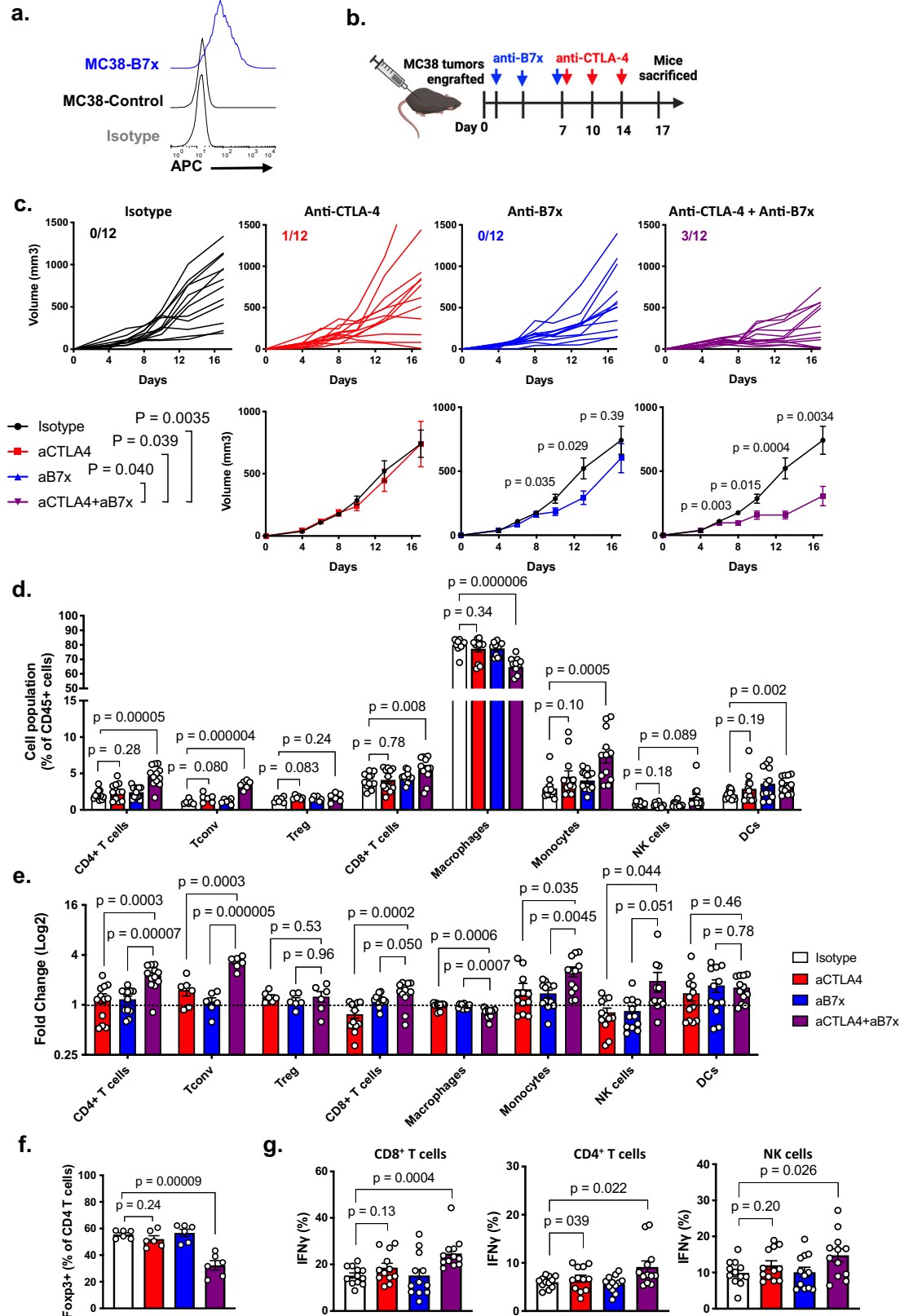

centrifuged in a discontinuous Percoll gradient: cells were resuspended 40% Percoll solution layered upon 80% Percoll, and then centrifuged at 320 g for 25 min. The cell-enriched interphase was extracted and used for downstream experiments.

**Cell culture**. The MC38 and CT26 cell lines were obtained from the Einstein Cytogenetics Facility cell line repository. The SKBR-3, MDA MB-468, OVCAR-3,

OVCAR-4, and NCI HC322M cell lines were derived from the National Cancer Institute Developmental Therapeutics Program cell line repository. The Hepa1-6 cell line was obtained from the Marion Bessin Liver Center at the Albert Einstein College of Medicine. MC38-derived cell lines were cultured in RPMI (Corning) supplemented with 10% FBS. Hepa1-6 and CT26-derived cell lines were cultured in DMEM (Corning) supplemented with 10% FBS. Primary T cells were cultured in T cell medium: RPMI supplemented with 10% FBS, 100 U/mL penicillin-

**Fig. 6 Combination anti-B7x and anti-CTLA-4 therapy has synergistic therapeutic effect. a** Representative staining of MC38-B7x and MC38-Control cells stained with anti-B7x clone 1H3. **b** Experimental scheme of MC38-B7x engraftment and anti-B7x or anti-CTLA-4 treatment. **c** Tumor volumes were tracked and shown as individual tracings (top), numbers designate tumor clearance (final tumor volume < 50 mm$^3$), and mean tumor volumes were calculated (bottom). **d, e** CD45$^+$ immune cells were quantified and calculated as proportion of total immune cell population (**d**), and fold change of treated groups were normalized to isotype-treated population (**e**). **f** Tregs were quantified. **g** Cell suspensions were stimulated with PMA/ionomycin and were stained intracellularly for IFN-γ expression. $n = 12$ mice per group, representative of three independent experiments. Error bars represent SEM, P values were calculated by two-tailed Student's T-test.

streptomycin, 10 mM HEPES, L-glutamine, 1 mM sodium pyruvate, MEM non-essential amino acids, and 55 μM β-mercaptoethanol. T cell medium was also supplemented with cytokines (Biolegend) as needed; 10 ng/mL rhIL-2, and variable concentration of rhTGFβ1 (Biolegend) as noted per experiment.

**Generation of stable cell lines**. The murine stem cell virus (MSCV) retroviral transduction system was used to generate B7x-expressing and B7x-negative control cell lines. Full length murine B7x was subcloned into the MSCV plasmid by Gibson Assembly (NEB) to generate MSCV-B7x, and the empty MSCV plasmid was used for Control cell lines. The MSCV-B7x or MSCV-Control plasmids were then cotransfected with pVSV-g packaging plasmid into Phoenix-Ampho packaging cells using jetPRIME transfection reagent (Polyplus) to produce virus. Virus-containing supernatant was then sterile-filtered, supplemented with polybrene and HEPES, and transferred to the tumor cells, which were then spinfected by centrifugation at 1000 g for 2 h at 37 °C. 4–7 days post-transduction, the transduced cells were flow-sorted using the BD FACSAria to reach >99% purity.

**In vitro Treg induction**. Purified splenic CD4$^+$ T cells were cultured in 96-well tissue culture-treated plates that were pre-treated with 3 μg/mL anti-CD3e and 2 μg/mL anti-CD28, as well as purified B7x-IgFc or control IgG2a-Fc protein (R&D). The plates were incubated for at least 2 h at 37 °C to allow the proteins to become plate-bound. The T cells were cultured in the coated plates in T cell medium supplemented with rhIL-2, and rhTGFβ1 was added 12–16 h after initial culture. The cells were harvested for analysis after 4 days of culture.

**Tumor Cell – T Cell coculture assay**. CD4$^+$ T cells were isolated as described above. MC38 or Hepa1-6 tumor cells were harvested from culture, along with the conditioned medium. The tumor cells were treated with 50 μg/mL mitomycin-c for 30 min at 37 °C. Tumor cells were then washed twice with medium, and co-cultured with T cells at a 1:2 ratio (50 K tumor cells to 100 K T cells) and 50 K anti-CD3/CD28 Dynabeads. The tumor conditioned medium was transferred back to the coculture, supplemented with HEPES, L-glutamine, MEM, b-mercaptoethanol, and IL-2. T cells were analyzed by flow cytometry after 4 days. In experiments with anti-TGFβ blockade, anti-TGFβ (Bioxcell) or isotype control was added into culture at 20 μg/mL.

**In vitro Treg suppression assay**. CD4$^+$ T cells were isolated from spleens of Foxp3-GFP-DTR mice and induced into iTregs with 1 ng/mL TGFβ1 as described above. After 4 days of Treg induction, resulting Foxp3-GFP$^+$ cells were flow-sorted, and CD4$^+$ or CD8$^+$ responder T cells were magnetically isolated from CD45.1 mice. Responder T cells were labeled with 1 μM CTV for 5 min. 40 K responder T cells were cultured with varying ratios of purified iTregs, as well as 40 K anti-CD3/anti-CD28 Dynabeads (Thermofisher). Proliferation of responder T cells was analyzed after 3 days.

**Small molecule inhibitors**. Chemical inhibitors were used to inhibit the following signaling pathways based on published literature. Akt – 1 μM GSK690693. Foxo – Foxo1 inhibition with 50 nM AS1842856 plus Foxo3 inhibition with 80 μM car-benoxolone. PTEN - 1 μM SF1670. Shp1 – 10 μM TPI-1. Shp2 – 10 μM Shp099. All compounds were obtained from Cayman Chemical.

**Immunofluorescence**. CD4$^+$ T cells cultured in 16-well Lab-Tek chamber slides (Thermofisher) were fixed with 4% paraformaldehyde for 20 min. Fixed cells were washed twice with PBS, and permeabilized with Perm buffer (0.1% Triton X-100 in PBS). Following fixation, cells were stained with anti-Foxo1 (CST) in Perm buffer for 30 min, washed twice, and then stained with anti-rabbit Alexa647 secondary antibody (Thermofisher) for 1 h. Cells were washed again with Perm buffer, and stained with Phalloidin Alexa488 (Thermofisher) for 1 h. Lastly, cells were washed a final time with PBS, and then mounted in DAPI Diamond Antifade (Thermofisher).

**Syngeneic tumor models**. Eight to ten-week-old female mice were used for syngeneic tumor grafts. At least 24 h prior to engraftment, the flanks and dorsal surfaces of the mice were shaved. Tumor cells were harvested from culture, resuspended in PBS, and injected subcutaneously into the flanks. $5 \times 10^5$ and $2 \times 10^6$ cells were used for MC38 and Hepa1-6 grafts in C57BL/6 mice, respectively,

and $5 \times 10^5$ cells were used for CT26 grafts in BALB/c mice. Tumor growth was measured by digital caliper, using the equation V = LxW$^2$/2 to calculate tumor volume. Tumors were allowed to grow until the humane endpoints (volume exceeding 2000 mm$^3$), or mice showing sick or moribund status, upon which the animals were sacrificed. Anti-CTLA-4 antibody treatments were administered on days 7, 10, and 14. Anti-B7x antibody treatments were administered on days 1, 4, 7, 10, and 14.

**Monoclonal antibodies**. Purified anti-CTLA-4 (clone 9D9), mouse IgG2b isotype (clone MPC-11) and anti-TGFβ (clone 1D11.16.8) were obtained from BioXCell. Anti-B7x was generated in-house from hybridoma culture. Hybridoma cells from clone 1H3 were cultured in Wheaton CELLine 350 flask. The cell compartment media was DMEM (Corning) supplemented with 10% ultra-low IgG FBS (Thermofisher), 10% NCTC-109 (Thermofisher), 1% non-essential amino acids, and 1% penicillin-streptomycin. The medium compartment media contained DMEM (Corning) supplemented with 1% penicillin-streptomycin. Antibodies were purified from cell culture supernatant by Protein G resin pulldown (Genscript). The purity and identity of antibodies were confirmed by SDS/PAGE.

**Tissue dissociation**. Tumors were chopped and incubated for 40 min, shaken continuously at 37 °C, in dissociation cocktail: 200 IU/mL Collagenase IV (Gibco), 0.5 IU/mL Dispase I, and 100 U/mL DNase I in serum-free medium. Dissociated cell suspensions were strained through 100 μm cell strainers, and subsequently centrifuged on a discontinuous 40–80% Percoll gradient. The resulting interphase was extracted and used for downstream experiments.

**Flow cytometry**. The following fluorophore-conjugated antibodies were used for cell surface staining (Supplementary Table 1a): anti-CD45, anti-CD3, anti-CD4, anti-CD8a, anti-CD25, anti-CD11b, anti-CD11c, anti- Ly6C, anti-Ly6G, anti-NK1.1, anti-Neuropilin-1, anti-F4/80, anti-TGF-LAP. For intracellular cytokine staining, cell suspensions were stimulated prior to staining with the PMA/iono-mycin-based Cell Stimulation Cocktail (Tonbo). Intracellular staining was performed with the eBioscience Transcription Factor Staining Buffer (Thermofisher) for the following antibodies (Supplementary Table 1b): anti-Foxp3, anti-IFN-γ, anti-Helios. Viability staining was performed with Ghost Dye Red 780 (Tonbo).

For phospho-flow cytometry of CD4$^+$ T cells, cells were removed from culture and immediately fixed with paraformaldehyde-containing fixation buffer (Biolegend), and then permeabilized with pre-chilled methanol-containing Permeabilization buffer (Biolegend) for 1 h at −20 °C. Cells were then stained for cell markers anti-CD4, anti-CD3, and a phospho-specific antibody (Supplementary Table 1c): anti-p-Smad2/3 (BD), anti-p-p65 (CST), anti-p-c-Jun (CST), anti-p-Akt (CST), anti-p-Foxo1 (CST), anti-p-STAT1 (CST), anti-p-STAT3 (Biolegend), anti-p-STAT4 (BD), or anti-p-STAT5 (BD). Unconjugated antibodies were detected with anti-Rabbit Alexa 647 secondary antibody (Thermofisher).

Fluorescence-minus-one or isotype-stain controls were used for all functional markers, phospo stains, and key populations such as Foxp3$^+$ cells.

For mitochondrial ROS staining, T cells were stained in culture with 5μM MitoSOX for 10 min at 37 °C. Subsequently, cells were washed, stained with anti-CD3, anti-CD4, viability dye, and analyzed by flow cytometry.

Stained samples were analyzed on the BD LSR-II or BD FACSCalibur, and subsequent data analysis was performed with Flowjo 10.7.

**Confocal microscopy**. Lab Tek 16-well chamber well slides (Thermofisher) were coated with anti-CD3e, anti-CD28, and B7x-Ig or Control-Ig as described in the Treg Induction protocol. CD4$^+$ T cells were grown in the wells for 24 h in T cell medium supplemented with IL-2 and 1 ng/mL TGFβ1 prior to analysis. The cells were fixed with 4% PFA for 20 min, followed by permeabilization with 0.1% Triton X-100. Cells were stained with anti-Foxo1 and anti-rabbit Alexa 647, Phalloidin Alexa 488 (Thermofisher), and then mounted with DAPI Diamond Antifade (Thermofisher).

Stained slides were analyzed on the Leica SP5 confocal microscope using the Leica LAS X acquisition software, and subsequent data analysis was performed on Fiji/ImageJ. For colocalization analysis, the Coloc2 tool ImageJ plugin was used on the DAPI and Foxo1-stained (Far Red) channels; Despeckle and Rolling Ball background subtraction was applied equally to both Control-Ig and B7x-Ig groups to improve signal-noise ratio on Foxo1 staining prior to Coloc2 analysis.

**Western blot**. CD4[+] T cells were grown in Treg-inducing conditions with either B7x-Ig or Control-Ig stimulation as described above in 6-well cell culture dishes. After 24 h, the medium was aspirated and the cells were lysed with RIPA buffer (1% NP-40, 0.5% deoxycholate, 0.1% SDS, 50 mM Tris, 150 mM NaCl) supplemented with Pierce protease and phosphatase inhibitor cocktail (Thermofisher). Total protein in the lysates were measured by BCA protein assay (Thermofisher). Equal total protein volumes were run on ExpressPlus 4–12% gels (Genscript) and wet-transferred to nitrocellulose membranes (Thermofisher). EZ-Run Prestained protein ladder (Fisher Scientific) was used for monitoring transfer efficiency and approximating molecular weights. The membranes were blocked with 5% milk in TBST for 1 h at room temperature, then probed with primary antibodies anti-Akt, anti-p-Akt, anti-Foxo1, anti-p-Foxo1 (CST) overnight at 4 °C, and then stained with secondary anti-rabbit HRP (CST) for 1 h at room temperature. Blots were detected with Pierce ECL substrate kit (Thermofisher) and visualized with the ChemiDoc imaging system (Bio-Rad).

**RNA isolation**. For in vivo tumor-infiltrating Treg analysis, Foxp3-GFP-DTR mice were engrafted with MC38-B7x or MC38-Control tumors, and tumors were dissociated as described above. 10,000 to 20,000 tumor-infiltrating Tregs, defined as CD45[+] CD4[+] GFP[+] cells, were flow-sorted on the BD FACSAria directly into RNA lysis buffer. For in vitro Treg analysis, splenic CD4[+] T cells from Foxp3-GFP-DTR mice were differentiated into iTregs with either B7x-Ig or Control-Ig, and subsequently GFP[+] cells were sorted directly intro RNA lysis buffer. RNA was subsequently extracted with the RNeasy Mini Kit (Qiagen) and stored at −80 °C. RNA quality of RIN ≥ 8.0 was confirmed by Bioanalyzer RNA Pico (Agilent) prior to sequencing. Library preparation and sequencing was performed by Admera Health. SMARTer Stranded Total RNA Pico (Takara) was used for library preparation, and sequencing was performed at 40 M PE reads per sample.

**RNA-Seq data analysis**. RNA-seq read were aligned to the mouse genome (mm10) using STAR (version 2.6.1b)[53]. The number of RNA-seq fragments mapped to each gene in the reference gene annotation (downloaded from the UCSC genome browser in 11/2019) was then counted using HTseq (version 0.6.1)[54]. Genes were considered expressed if their average expression counts were greater than or equal to 1 in any of the two sample groups, and selected for deferential expression analysis and principal component analysis (PCA) by DESeq2 (version 3.11)[55]. The RNA-seq data are deposited in the Gene Expression Omnibus (GSE199751). Gene sets for GSEA analysis were taken from "Treg Spleen Effector" and "Treg Colon Suppressive" published gene data sets[29], analyzed with GSEA 4.1.0 (Broad Institute)[56]. For TCGA analyses of B7x and Foxp3 transcriptomic correlation, mRNA expression RSEM data was accessed through cBioPortal and analyzed in Graphpad Prism 8; only samples with RSEM gene expression values of >0 were used for the analysis. The following data sets from the TCGA PanCancer Atlas were used: Colorectal adenocarcinoma (COAD), breast invasive carcinoma (BRCA), sarcoma (SARC), low grade glioma (LGG), prostate adenocarcinoma (PRAD), head and neck squamous cell carcinoma (HNSC), thymoma (THYM), kidney chromophobe (KICH), skin cutaneous melanoma (SKCM), liver hepatocellular carcinoma (LIHC), thyroid carcinoma (THCA), glioblastoma (GBM), and uterine carcinosarcoma (UCS). Correlations between B7x and the Treg gene signature were performed through the TIMER 2.0 webserver[57], quantified by the quanTIseq algorithm[28].

**Statistics**. Data were analyzed with GraphPad Prism 8 software. Statistical significance was determined by unpaired Student's t-test or paired t-test (with equal variation assumption unless F test indicated unequal variations). Details for each experiment such as sample size n and specific tests are provided in corresponding figure legends. Data in bar graphs are shown as mean values with +/− standard error of mean (SEM), and symbols represent replicates. In general, replicates for in vivo experiments indicate biological replicates, i.e., samples from individual mice, and replicates for in vitro experiments indicate technical replicates. In vivo experiments were repeated at least twice following a preliminary experiment, in vitro experiments were repeated three or more times, as specified in the figure legends.

**Reporting summary**. Further information on research design is available in the Nature Research Reporting Summary linked to this article.

## Data availability
The sequencing data that support findings in this study are deposited in the Gene Expression Omnibus with accession code GSE199751. All TCGA datasets used in this paper were accessed from the TCGA PanCancer Atlas, accessed through cBioPortal (https://www.cbioportal.org), and are also provided in the Source Data file. The remaining data are available within the Article, Supplementary Information, or Source Data file.

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

## Acknowledgements

P.J. is supported by National Institutes of Health (NIH) F30CA224931. P.M.G. and M.C.P. are supported by NIH 5TL1TR002557. This work was supported by NIH 3R01CA175495-09S1 and Department of Defense BC190403 (to X.Z.).

## Author contributions

Conceptualization: P.J. and X.Z.; investigation: P.J.; methodology: P.J. and K.C.O.; formal analysis: P.M.G. and D.Z.; resources: M.C.P. and Y.W.; writing–original draft: P.J.; writing–review and editing: X.Z., P.M.G., M.C.P., Y.W., K.C.O., and D.Z.; supervision: X.Z.; funding acquisition: X.Z.

## Competing interests

X.Z. is an inventor on patent 9447186 covering cancer immunotherapy targeting B7x. P.J., M.C.P., Y.W., and X.Z. are inventors on a pending patent (Novel anti-B7x antibodies and derivative products). Other authors declare no conflicts of interest

## Additional information

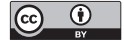

**15**