## [Peer Review File · Nature Communications]

Reviewers' Comments:

Reviewer #3:

Remarks to the Author:

The weakness of this paper is that the model is artificial. There may be some human tumours that express B7x but this is not apparently the case for mouse tumours. We have no idea what the reason for this is (other receptors etc) but there seems no mouse tumour that normally expressed B7x.

They should have considered a human tumour model in an immunocompromised mouse and reconstitute the human immune system by cell transfer, a technique commonly used.

They must state this issue in the abstract and text so as not to be misleading. a statement such as 'although known mouse tumors do not express B7x, we used tumors transfected to express B7x in order to study effects on tumor immunotherapy'

Regarding responsive vs non-responsive, they authors do not seem to understand. Of course there is variability but as in any tumour model, there are mice that are more responsive than others and hence can be separated into responsive vs non-responsive..for example >50% reduction is positive..why not here as in many papers?

In reference to CD80/86, I am surprised since many tumors express some CD80/86.

The anti-TGFbeta effects are nice.

Reviewer #4:

Remarks to the Author:

In this revised manuscript, the authors have mostly responded to the points raised in the previous comments by Reviewer #2. However, there are a few issues remaining in the rebuttal. In addition, there are several issues that have not yet been raised by Reviewer #2 or #3.

1. Comment 4 and Comment 6.4 are either missing or mislabeled.
2. Regarding to Comment 6.1, CTLA-4 expression by FACS in Figure 5 do not show clear separation between CTLA-4 negative and positive cells. The authors should determine MFI of CTLA-4 expression to support that CTLA-4 expression is reduced by B7x.
3. Related to Comment 6.1, the authors state that "B7x-mediated Treg expansion interferes with the Treg-depleting effects of anti-CTLA-4 (Fig. 5)" in the Discussion section. However, Figure 5 suggests that B7x-mediated reduction in CTLA-4 expression by Treg cells impairs depletion of Treg cells by anti-CTLA-4, which requires clarification.
4. In Figure 4, whether B7x decreased TCR signaling is unclear. Since the receptor for B7x is unknown, the authors should specifically address whether B7x affects TCR signaling induced by TCR stimulation alone rather than iTreg-inducing condition.
5. Foxp3 PE-Cy5.5 and CD25 PerCP-Cy5.5 were used to determine the frequency of Treg cells, however, these two dyes have major overlaps in their emission spectra. The authors should indicate the instrument setting and clarify that the gated population represents Treg cells.

6. In Figure 1p, TCGA study abbreviation alone might be confusing for the readers.

The authors thank the Editor for the positive editorial comment. We would like to express our sincere gratitude for giving us the chance to resubmit our paper. We appreciate the editor and the reviewers for the time they have spent to evaluate this manuscript. We have addressed all the issues that were raised. In addition, we have clarified all the points suggested by the reviewers and have appropriately made the corrections. A point-by-point response to the reviewers' comments are described below. Please note that new modifications are highlighted in yellow in the revised manuscript.

Reviewer #3 (Remarks to the Author):

Comment 1: The weakness of this paper is that the model is artificial. There may be some human tumours that express B7x but this is not apparently the case for mouse tumours. We have no idea what the reason for this is (other receptors etc) but there seems no mouse tumour that normally expressed B7x.

Response: We acknowledge the reviewer's observations regarding the model. B7x not being expressed by murine tumor cell lines is an obstacle to studying the biology of this important immune checkpoint orphan ligand. However, in our studies, we showed that the stably transduced cell lines have B7x expression levels comparable to naturally-expressing human tumor cell lines, and that the stably transduced model has consistent physiological effects across multiple cell lines.

Comment 2: They should have considered a human tumour model in an immunocomprised mouse and reconstitute the human immune system by cell transfer, a technique commonly used.

Response: Given there are advantages and disadvantages with every animal tumor model, we felt that syngeneic tumor grafts would allow us to perform detailed mechanistic studies in a fully immunocompetent model. We have experience on humanized mouse models [Sci Immunol. 2021 Jul 9;6(61):eabf9792], however, current humanized mouse models are not suitable for the study of this manuscript. In order to study how B7x promotes tumor-infiltrating Tregs and resistance to anti-CTLA-4 therapy, a humanized mouse model has to have the exact same MHCs of human tumor cells expressing endogenous B7x, antigen-presenting cells, naïve T cell cells, and Tregs, but currently there is no such a humanized mouse model available.

Comment 3: They must state this issue in the abstract and text so as not to be misleading. a statement such as 'although known mouse tumors do not express B7x, we used tumors transfected to express B7x in order to study effects on tumor immunotherapy'

Response: Previously, our main text already included the following, "Whereas human tumor cell lines express B7x at high levels, murine cell lines commonly used in syngeneic tumor models do not natively express B7x (Extended Data Fig. 1a), therefore we transduced the well-characterized murine tumor cell lines MC38, CT26, and Hepa1-6 to express mouse B7x. These stably transduced cell lines express B7x at levels comparable to human cell lines that natively express human B7x (Extended Data Fig. 1b)." (page 3, lines 90-94). As suggested by the reviewer, we have now included clarification in the abstract.

Modification: Please see the following modification of the abstract: "Here, we show that transduction and stable expression of B7x in multiple syngeneic tumor models leads to the expansion of immunosuppressive regulatory T cells (Tregs)" (page 1, lines 29-31)."

Comment 4: Regarding responsive vs non-responsive, they authors do not seem to understand. Of course there is variability but as in any tumour model, there are mice that are more responsive than others and hence can be separated into responsive vs non-responsive..for example >50% reduction is positive..why not here as in many papers?

Response: We agree with the reviewer that stratifying data into responders vs. non-responders is helpful, particularly for data that shows a bimodal distribution. For this reason, in experiments where mice were given pharmacological treatments, we quantified responders vs. non-responders based on pre-determined tumor volume thresholds (Fig. 5d and 6c). An example is shown below (red arrows):

Since the mice in the experiments of Fig. 1 were not given treatments, we avoided using the term “responder” to avoid confusion between the different types of experiments. Moreover, we expected data in Fig 1 to follow unimodal normal distributions, although as the reviewer points out, sometimes the data does not neatly fall into this pattern. Therefore, in an effort to provide a binary quantification to the experiments in Fig.1 as per the reviewer’s recommendation, we have displayed the fraction of mice for which the B7x tumors show an increase in the respective measurement relative to the Control tumors. An example is shown below (red arrow):

Modification: Please find graphs corresponding to the paired tumor experiments (Fig. 1j, k, m, n) to be updated with fractions as described above. In addition, the following text is included in the Figure legend: “...fraction of mice for which the B7x tumors show an increase in the respective measurement relative to the Control tumors are displayed in corners of graph” (page 18, lines 775-777).

Comment 5: In reference to CD80/86, I am surprised since many tumors express some CD80/86.

Response: We appreciate the reviewer’s insight that some tumor cell lines do express CD80/CD86, although the tumor cells used here evidently do not express either ligand at detectable levels.

Comment 6: The anti-TGFbeta effects are nice.

Response: We appreciate the reviewer's positive comments and agree that the TGF-blockade data support the conclusion that the tumor cells used in these models induce iTreg conversion in a TGF β -dependent manner.

Reviewer #4 (Remarks to the Author): to replace original Reviewer #2

In this revised manuscript, the authors have mostly responded to the points raised in the previous comments by Reviewer #2. However, there are a few issues remaining in the rebuttal. In addition, there are several issues that have not yet been raised by Reviewer #2 or #3.

Comment 1: Comment 4 and Comment 6.4 are either missing or mislabeled.

Response: We thank the reviewer for identifying this typographic error. The numbers were mislabeled, no information was missing in the prior document.

Comment 2: Regarding to Comment 6.1, CTLA-4 expression by FACS in Figure 5 do not show clear separation between CTLA-4 negative and positive cells. The authors should determine MFI of CTLA-4 expression to support that CTLA-4 expression is reduced by B7x.

Response: As the reviewer suggests, we quantified MFI in addition to percent positive and have included this data into the manuscript. Both methods of quantification show the same result.

Modification: Please find an additional graph for Fig. 5b quantifying CTLA-4 MFI, and additional text in the Figure legends: "... graphs quantifying percent CTLA-4+ of total Tregs and CTLA-4 MFI of Tregs are shown" (page 19, 834-835).

Comment 3: Related to Comment 6.1, the authors state that "B7x-mediated Treg expansion interferes with the Treg-depleting effects of anti-CTLA-4 (Fig. 5)" in the Discussion section. However, Figure 5 suggests that B7x-mediated reduction in CTLA-4 expression by Treg cells impairs depletion of Treg cells by anti-CTLA-4, which requires clarification.

Response: We thank the reviewer for identifying this potential source of confusion and have edited the text in the Discussion section to improve clarity.

Modification: Please find the following modification in the main text, "Expression of B7x in tumors reduces the efficacy of anti-CTLA-4 therapy by impairing Treg depletion (Fig. 5)" (page 8, lines 360-361).

Comment 4: In Figure 4, whether B7x decreased TCR signaling is unclear. Since the receptor for B7x is unknown, the authors should specifically address whether B7x affects TCR signaling induced by TCR stimulation alone rather than iTreg-inducing condition.

Response: Regarding the first point, we would like to clarify that it is highly likely that B7x inhibits TCR signaling based on the data in our manuscript (Fig. 5). In addition, we previously reported that B7x inhibited cytokine production and proliferation of anti-CD3-activated T cells [PNAS 2003 Sep 2;100(18):10388-92]. Moreover, TCR signaling is a key aspect of CD4+ T cell differentiation. Indeed, TCR stimulation via anti-CD3 is a necessary component of Treg-inducing protocol used in our manuscript, as is standard protocol. Thus, in all phospho-flow and phospho-westerns experiments in this manuscript, Treg-inducing conditions containing anti-CD3 stimulation were used. We previously highlighted this in the main text with the following, “we first analyzed the phosphorylation status of transcription factors downstream of the T cell receptor (TCR) complex in CD4+ T cells cultured in Treg-inducing conditions” (page 5, lines 206-208). As suggested by the reviewer, we now added further clarification in the text.

Modification: Please find the following modification, “Collectively, our data demonstrate that B7x inhibits TCR signaling in Treg-inducing conditions which subsequently suppresses Akt and activates Foxo1 to promote Foxp3 expression.” (page 6, lines 260-261).

Comment 5: Foxp3 PE-Cy5.5 and CD25 PerCP-Cy5.5 were used to determine the frequency of Treg cells, however, these two dyes have major overlaps in their emission spectra. The authors should indicate the instrument setting and clarify that the gated population represents Treg cells.

Response: Whereas fluorescence spillover is certainly a concern for multicolor flow cytometry, we were careful to use FMO or isotype stain controls to determine gating in all flow cytometry experiments used in this manuscript. For both Foxp3 and CD25 gating, FMOs were used to confirm positive vs. negative populations, and we observed minimal spillover after appropriate fluorescence compensation (see below).

Gated on CD4+ T cells:

Modification: In the methods section, the following text has been included: “Fluorescence-minus-one or isotype-stain controls were used for all functional markers, phosfo-stains, and key populations such as Foxp3+ cells.” (page 12, lines 522-523).

Comment 6: In Figure 1p, TCGA study abbreviation alone might be confusing for the readers.

Response: To avoid confusion, further clarification has been provided in the Methods section (see below).

Modification: Please find the following text in the Methods section, “The following data sets were used: Colorectal adenocarcinoma (COAD), breast invasive carcinoma (BRCA), sarcoma (SARC), low grade glioma (LGG), prostate adenocarcinoma (PRAD), head and neck squamous

cell carcinoma (HNSC), thymoma (THYM), kidney chromophobe (KICH), skin cutaneous melanoma (SKCM), liver hepatocellular carcinoma (LIHC), thyroid carcinoma (THCA), glioblastoma (GBM), and uterine carcinosarcoma (UCS).” (page 13, lines 567-571).

Reviewers' Comments:

Reviewer #4:

Remarks to the Author:

In the revised manuscript, the authors have adequately responded to the points raised in the comments.